# Synthesis of Co_3_O_4_ Nanoplates by Thermal Decomposition for the Colorimetric Detection of Dopamine

**DOI:** 10.3390/nano12172990

**Published:** 2022-08-29

**Authors:** Zengmin Tang, Ling Zhang, Sijia Tang, Junping Li, Jianxiong Xu, Na Li, Lijian Xu, Jingjing Du

**Affiliations:** 1Hunan Key Laboratory of Biomedical Nanomaterials and Devices, College of Life Sciences and Chemistry, Hunan University of Technology, Zhuzhou 412007, China; 2Yichun Fangke Sewage Treatment Co., Ltd., Mingyue North Road 542, Yichun 336000, China; 3Hunan Key Laboratory of Electrochemical Green Metallurgy Technology, College of Materials and Advanced Manufacturing, Hunan University of Technology, Zhuzhou 412007, China; 4College of Packaging and Materials Engineering, Hunan University of Technology, Zhuzhou 412007, China

**Keywords:** Co_3_O_4_, thermal decomposition, oxidase-like activity, colorimetric detection, dopamine

## Abstract

Inorganic nanomaterials with enzyme-like activity have been attracting much attention due to their low cost, favorable stability, convenient storage, and simple preparation. Herein, Co_3_O_4_ nanoplates with a uniform nanostructure were prepared by the thermolysis of cobalt hydroxide at different temperatures, and the influence of the annealing temperature on the performance of the mimetic enzyme also was reported for the first time. The results demonstrated that Co_3_O_4_ nanoplates obtained at an annealing temperature of 200 °C possessed strong oxidase activity and efficiently catalyzed the oxidation of 3,3′,5,5′-tetramethylbenzidine (TMB) without the addition of hydrogen peroxide to generate the blue color product ox-TMB. Once the annealing temperature was increased to 500 °C and 800 °C, the oxidase activity of Co_3_O_4_ decreased rapidly, and was even inactivated. This might be attributed to the relatively large specific surface area of Co_3_O_4_ annealed at 200 °C. Besides this, based on the TMB-Co_3_O_4_ nanoplate system, a colorimetric analysis method was developed to detect dopamine with a limit of 0.82 μmol/L in a linear range from 1.6 μmol/L to 20 μmol/L.

## 1. Introduction

Dopamine is a well-known considerable catecholamine neurotransmitter in the human central nervous, cardiovascular, and hormone systems [1]. A deficient level of dopamine may cause Parkinson’s disease, and an excessive level of dopamine also may bring about abnormalities including schizophrenia and euphoria, etc. [2,3]. For the sake of convenience, it is necessary to develop point-of-care testing (POCT) for dopamine detection. The POCT system is an in vitro monitoring platform with a short turnout time; it can even can be operated by an unskilled user, which can promote the development of clinical testing and bring significant economic benefits [4]. Recently, various methods have been applied in POCT, such as fluorescence sensors, electrochemical methods, and colorimetric sensors [5,6,7,8,9,10,11].

In recent decades, electrochemical sensors to detect dopamine have received wide attention due to their facile operation and response [12,13,14]. However, the actual test experiments found that the research on the stability and reproducibility of the sensors still faces significant challenges. In addition, the recently developed colorimetric sensors based on oxidase-like nano-enzyme materials present a change in color and absorbance during the process of detecting dopamine, which also applies to the POCT system due to its low cost, simplicity, rapid measurement, and sensitive readout [15,16,17]. Compared with natural enzymes, nano-enzyme materials possess the advantage of high stability. The oxidase mimic can more effectively oxidize the colorless TMB into ox-TMB with a blue color in the absence of hydrogen peroxide [18]. The addition of dopamine can absorb the dissolved oxygen or reactive oxygen, eventually leading to the fading of the color and the reduction of absorbance. Recently, precious metal nanomaterials have been considered as promising oxidase mimic candidates for application in colorimetric sensors [19], such as gold nanoparticles [20], platinum nanoparticles [21], silver nanoparticles [22], gold–platinum alloy nanoparticles [23], and silver–platinum alloy nanoparticles [24], etc. However, their high cost and limited resources inhibit their development and wide application. Therefore, transitional metal oxides performing comparable oxidase-like activity have also received considerable attention as a low-cost and available alternative [25,26,27].

As a transitional oxide, Co_3_O_4_ is an important p-type semiconductor with wide applications in supercapacitors, batteries, and catalysis, etc. [28,29,30,31,32]. In addition, Co_3_O_4_ nanoparticles also exhibit the behavior of mimic enzymes, and have received much attention for colorimetric sensors, enzyme-mimicking catalyzers, and so on [33,34]. In 2012, Mu et al. first reported Co_3_O_4_ with a catalase-like activity, which was applied to detect the concentration of H_2_O_2_ [35]. Shortly afterward, Qin et al., in 2014, found that the synthesized Co_3_O_4_ showed oxidase-like activity and could oxidize TMB to form colored ox-TMB, which has been applied for the detection of sulfite in food [36]. These results also demonstrated the great potential of Co_3_O_4_ for the detection of dopamine [37].

Two-dimensional nanoparticles are particles with one dimension that is limited to the nanometer scale; they have received wide attention due to their unique shape, anisotropic physical and chemical properties, and high surface-to-volume ratios [38]. In this study, the Co_3_O_4_ nanoplates were synthesized by the thermal decomposition of the as-prepared precursor Co(OH)_2_ at different temperatures for 30 min. The mimic enzyme behaviors of Co_3_O_4_ obtained at different annealing temperatures were discussed. The oxidase activity of Co_3_O_4_ was evaluated by Michaelis–Menten. Eventually, performances such as the sensitivity, selectivity, and lowest detection limit in the detection of dopamine were investigated.

## 2. Chemical and Methods

The cobalt chloride hexahydrate (CoCl_2_ 6H_2_O), glacial acetic acid (CH_3_COOH), sodium acetate (CH_3_COONa), branched polyethyleneimine solution (BPEI, Mw: 60,000, 50 wt% in water), octylamine (C_8_H_19_N), ethanol (C_2_H_5_OH), 3,3′,5,5′-tetramethylbenzidine (TMB), potassium chloride (KCl), magnesium chloride (MgCl_2_), ferric chloride(FeCl_3_), and cupric sulfate (CuSO_4_) were obtained from InnoChem science and technology Co. Ltd. (Beijing, China).

Regarding the fabrication of the Co_3_O_4_ nanoplates, Co_3_O_4_ nanoplates were prepared through two steps. First, Co(OH)_2_ was synthesized using the hydrothermal method, as described in the previous literature [39]. Briefly, 2 mmol CoCl_2_ was added to 10 mL distilled water in the presence of 0.06 g BPEI, and was agitated constantly at 90 °C in a water bath. Then, the reaction was caused by adding 1 mL octylamine, and was maintained for around 40 min. After cooling, washing, and separation, the precursor Co(OH)_2_ was obtained. The second step was the decomposition of Co(OH)_2_ at different temperatures (200, 500, and 800 °C) for 30 min in a tube furnace in order to fabricate Co_3_O_4_ nanoplates. The samples synthesized at different calcination temperatures were named Co_3_O_4_-200, Co_3_O_4_-500, and Co_3_O_4_-800.

Regarding the catalytic activity of the Co_3_O_4_ and kinetic analysis, briefly, 0.1 mL of various Co_3_O_4_ suspensions with a concentration of 3 mg/mL and 0.2 mL TMB with a concentration of 2 mmol/L were added to 2.7 mL CH_3_COOH-CH_3_COONa buffer solution (0.2 mol/L, pH = 4.0), and then the mixture was incubated at 40 °C for 30 min. In this system, the concentration of Co_3_O_4_ was 100 μg/mL, and the concentration of TMB was 133.3 μmol/L. Then, the ultraviolet–visible absorptions of the above solution between the range of 550–750 nm were recorded. The kinetic parameters were conducted at 40 °C using different concentrations of TMB whilst keeping the other conditions constant. Kinetic analysis was performed by recording the absorbance at 652 nm after incubation for 30 min at 40 °C. The corresponding Michaelis–Menten constant was calculated by a Lineweaver-Burk plot [37]:1/υ = K_m_/V_m_ (1/[C] + 1/K_m_)
where K_m_ is the Michaelis–Menten constant, which is an indicator of Co_3_O_4_-200’s affinity for its TMB, [C] is the concentration of TMB, υ is the initial velocity, and V_m_ represents the maximal reaction velocity.

Regarding the colorimetric detection of dopamine, for the dopamine analysis, the absorption spectra of mixtures containing Co_3_O_4_-200 (0.1 mL, 3 mg/mL), CH_3_COOH–CH_3_COONa buffer solution (2.7 mL, 0.2 mol/L), TMB (0.2 mL, 2 mmol/L), and dopamine with different concentrations from 1.6 to 20 μmol/L were recorded from 550 to 750 nm. The change of absorbance (△A = A_0_ − A, where A_0_ and A are the absorbances of the solution without and within the dopamine sample) at 652 nm versus the concentration of dopamine was applied to obtain the calibration curve. For the determation of the anti-interference performance, dopamine was detected in the presence of KCl, MgCl_2_, FeCl_3_, CuSO_4_, H_2_O_2_, glucose, and urea with a concentration of 100 μmol/L.

Regarding the characterization, the morphology of the samples was characterized using a scanning electron microscope (SEM, Zeiss Gemini 300, Jena, Germany). The crystal structure was analyzed by X-ray diffraction (XRD, Bruker D8, Karlsruhe, Germany). The absorption spectrum in the range of 550–750 nm was measured using a UV-vis spectrophotometer (TU-1901). The surface area was detected by BET (Nova, Quantachrome, Boynton Beach, FL, USA).

## 3. Results and Discussion

### 3.1. Fabrication of Co_3_O_4_ Nanoplates

Based on the previous synthetic strategy [39], the pre-prepared Co(OH)_2_ nanoplates were first prepared and employed as the precursor to fabricate the Co_3_O_4_ nanoplates by regulating the annealing temperature. The thermal decomposition reaction only took 30 min at the designed temperature. The surface structure and oxidase-mimic activity of the Co_3_O_4_ nanoplates were significantly affected by the annealing temperature, as shown in Figure 1.

Powder XRD was used to characterize the crystal structure of the samples. Compared with the XRD pattern of Co(OH)_2_ (Figure 1a), the XRD pattern of Co_3_O_4_-200 in Figure 1b changes with the intensity of the peaks and the new diffraction peaks. Several weak diffraction peaks at 2θ = 19.0°, 31.3°, 36.8°, 44.8°, 59.3°, and 65.2° were assigned to the (111), (220), (311), (400), (511), and (440) crystal facets of cubic phase Co_3_O_4_, respectively. Besides this, the crystal structures of the products are not affected by the higher calcination temperatures, such as 500 °C and 800 °C. Therefore, calcination at 200 °C for 30 min can achieve the transformation of Co_3_O_4_ from Co(OH)_2_, which is 20 times shorter than the calcination time reported in the previous research [40].

SEM was applied to observe the morphology of samples obtained at the different annealing temperatures. The SEM image in Figure 2a confirms that the precursor Co(OH)_2_ presents a hexagonal plate shape with a smooth surface, which is consistent with the previous study [39]. After the thermal decomposition of Co(OH)_2_ nanoplates at 200 °C for 30 min, a black powder was obtained. As shown in the SEM image in Figure 2b, the obtained Co_3_O_4_-200 remains in the shape of a hexagonal plate with a size range of 2–15 μm. Compared with Co(OH)_2_, cracks on the surface of the black samples can be observed due to the evaporation of water produced in the process of the thermal decomposition of Co(OH)_2_. In addition, the cracks are supposed to increase the surface area, and should eventually enhance the adsorption capacity and catalytic capacity. The surface structure of the samples hardly changed at the temperature of 500 °C (Figure 2c). As the temperature increased to 800 °C, pore structures were produced due to the fusion and recrystallization of the samples in the process at a high calcination temperature (Figure 2d). As shown in Table 1, the Co_3_O_4_-200 nanoplates have a relatively large specific surface area, which is 15 times greater than the specific surface area of the Co_3_O_4_-500 and Co_3_O_4_-800. The Co_3_O_4_-800 presented a macropore structure, and the process of fusion and recrystallization would cause agglomeration, as shown in the insert SEM image in Figure 2d. In addition, the distribution of pore and pore size also was not uniform. Therefore, the sample Co_3_O_4_-800 has a lower surface area according to the comprehensive analysis combined with agglomeration and uneven macropores.

### 3.2. Oxidase-like Activity of the Co_3_O_4_ Nanoplates

The oxidase-like activities of various types of Co_3_O_4_ were investigated by the catalytic oxidation of TMB in the absence of hydrogen peroxide, and the corresponding absorption at the range of wavelength from 550–750 nm was measured after incubation at 40 °C for 30 min. The results are shown in Figure 3a. Curve 1 demonstrates that the direct oxidation of TMB by dissolved oxygen in the absence of Co_3_O_4_ can be ignored. The obtained Co_3_O_4_-200 nanoplates can catalyze the oxidation of colorless TMB to produce a blue product (ox-TMB), which has an obvious absorption peak at 652 nm (curve 4). On the contrary, the very weak absorption peaks in curve 2 and curve 3 reveal the sharply weakened oxidase enzyme behavior of the Co_3_O_4_-500 and Co_3_O_4_-800 nanoplates. Therefore, the oxidase activity of the obtained Co_3_O_4_ is seriously affected by the annealing temperature. In the process of catalysis, through adsorption for dissolved oxygen on the surface, the Co_3_O_4_ nanoplates are capable of catalyzing the dissolved oxygen to release reactive oxygen species accompanied by the oxidation of Co^2+^ to Co^3+^ [41]. In various reactive oxygen species, superoxide anions (O_2_^−^) acted as the main reactive oxygen species for the oxidation of TMB to blue ox-TMB. Due to its loose surface structure and larger surface area, Co_3_O_4_-200 nanoplates can adsorb and catalyze more dissolved oxygen to release more superoxide anions, showing stronger catalytic activity for the oxidation of TMB. The curve in Figure 3b demonstrates that the oxidation of TMB catalyzed by Co_3_O_4_-200 needs to take around 24 min to enter the steady state.

In order to better understand the Co_3_O_4_ nanoplates’ performance of an oxidase-like activity, the kinetic parameters of TMB alone were determined by varying the concentration of TMB, keeping the concentration of Co_3_O_4_ constant at 3 mg/mL. The absorbance change versus the concentration of TMB at 652 nm for 30 min is shown in Figure 3c. A typical Michaelis–Menten curve is shown in Figure 3d. The Michaelis–Menten constant (K_m_) and maximum initial velocity (V_m_) for the Co_3_O_4_-200 nanoplate with TMB are 0.058 mmol/L and 1.27 μmol/(L·s), which are greater than the kinetic parameters reported for Co_3_O_4_, Co_3_O_4_/CuO, and CoOOH [35,37,42]. It is suggested that synthesized Co_3_O_4_-200 nanoplates possess a high affinity for TMB.

Additionally, the effects of temperature and pH on the oxidase-like activity of Co_3_O_4_ were explored. Firstly, the TMB-Co_3_O_4_-200 suspension was incubated for 30 min at different temperatures while the pH was maintained at 4.0. The absorbance was recorded, and is shown in Figure 4a,b, which shows a similar absorbance in the range of 20 to 60 °C and a decrease of absorbance as the temperature increases to 70 °C. Then, the pH of the CH_3_COOH-CH_3_COONa buffer solution was controlled at 3.6, 4.0, 4.4, 4.8, and 5.2 as the temperature was kept at 40 °C. Figure 4c,d demonstrates that Co_3_O_4_ nanoplates possess a good oxidase-like activity at a pH of 3.6–4.4. The absorbance decreases once the pH rises to 4.8 and 5.2. Therefore, the results illustrate that the Co_3_O_4_ can maintain a high oxidase-like activity in a wide temperature range from 20 to 60 °C with a pH of around 4.0.

### 3.3. Detection of Dopamine Employing the Co_3_O_4_ Nanoplates

As we found above, Co_3_O_4_-200 can be used as an oxidase mimic, and can catalyze TMB to produce colored ox-TMB without the presence of hydrogen peroxide. In this research, in order to investigate the applicability of the oxidase-like Co_3_O_4_-200, colorimetric detection for dopamine was conducted due to its inhibiting effect on the oxidase-like Co_3_O_4_-200 nanoplates. Figure 5a shows that the blue color gradually fades with the increasing concentration of dopamine. Figure 5b further shows that the absorbance of the Co_3_O_4_-TMB mixtures at 652 nm also gradually reduce with the increase of the dopamine concentration from 1.6 μmol/L to 25 μmol/L, and a further increase in the concentration of dopamine only has a minor change in absorbance. The fading of the blue demonstrates that the oxidase-like mimic of Co_3_O_4_-200 is inhibited by dopamine. This might be attributed to the fact that dopamine can induce the reduction of ox-TMB, or that dopamine more easily consumes the dissolved oxygen, which eventually leads to the decrease of the amounts of reactive oxygen species [43]. The corresponding ΔA versus the concentration of dopamine is shown in Figure 5c, which demonstrates that the concentration of detectible dopamine is around 1.6 μmol/L in a linear range from 1.6 μmol/L to 20 μmol/L. The limit of detection (LOD) for dopamine is as low as 0.82 μmol/L (S/N = 3).

### 3.4. Analytic Performance for the Selectivity of Dopamine Detection

Selectivity is an important property of mimetic enzymes. Figure 6 shows the selectivity of oxidase Co_3_O_4_-200 for dopamine detection in the presence of various substances, including K^+^, Mg^2+^, Fe^3+^, Cu^2+^, Cl^−^, SO_4_^2−^, H_2_O_2_, glucose, and urea. The absorbance of the TMB-buffer-Co_3_O_4_-200 system at 652 nm shows a significant decrease in the presence of 10 μmol/L dopamine, while there are no significant changes in the absorbance for other substances with a concentration of 50 μmol/L. These results indicate that the oxidase Co_3_O_4_-200 displays good selectivity for the detection of dopamine.

## 4. Conclusions

Herein, Co_3_O_4_ nanoplates synthesized by thermal deposition possessed oxidase activity. The annealing temperature played an important role in tuning the oxidase activity of Co_3_O_4_ nanoplates. In these samples, Co_3_O_4_ nanoplates obtained at an annealing temperature of 200 °C showed excellent oxidase activity due to their relatively large specific surface area and loose structure. As oxidase mimics, Co_3_O_4_ nanoplates were employed to develop a colorimetric sensor of dopamine with a limit of 0.82 μmol/L in a linear range from 1.6 μmol/L to 20 μmol/L.

## Data Availability

Not applicable.

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
