# Peer review of "Synthesis of Co3O4 Nanoplates by Thermal Decomposition for the Colorimetric Detection of Dopamine"

_nanomaterials, 2022, doi:10.3390/nano12172990_

Round 1

Reviewer 1 Report

The authors report that the synthetic method to produce Co3O4 nanoplates and their applications in the field of sensing technologies. The manuscript is important in the topic of sensing technology with high selectivity and I think this manuscript can be publishable in Nanomaterials. However, it needs to be revised for the following issues before the publication.

1. There is a typo (subscript, Page #6) to be revised. I would recommend that authors should double check their grammar and expressions in English.

2. The enzymatic properties of Co3O4 nanoplates should additionally explained (e.g. the mechanism of oxidase-like properties). In addition, the importance of Co3O4 nanoplates in colorimetric sensing technologies should be addressed in detail.

3. In Figure 2, SEM images of Co3O4 nanoplates seems to be more porous structures with increasing calcination temperature. Author should address the reason why their specific surface areas were decreased at higher calcination temperatures even though they have more porous structures.

4. According to the calcination temperature, the oxidase activities of Co3O4 nanoplates were seriously decreased. Authors should explain this phenomenon and mechanism as well.

Author Response

Dear Reviewer:

Many thanks for the comments dated Aug. 19th, 2022 on the manuscript “Synthesis of Co3O4 nanoplates by thermal decomposition for colorimetric detection of dopamine”. We have carefully considered the reviewer’s comments and revised our manuscript accordingly. The changes are highlighted with a yellow pen in the revised manuscript. Here we would like to address the editor’s comments:

Comment 1. There is a typo (subscript, Page #6) to be revised. I would recommend that authors should double check their grammar and expressions in English.

Response 1: We would like to thank this reviewer’s kind suggestion. Several typos have been corrected and the expressions and grammar also have been modified in this revised manuscript after double check the whole text. The modification has been highlighted.

Comment 2. The enzymatic properties of Co3O4 nanoplates should additionally explained (e.g. the mechanism of oxidase-like properties). In addition, the importance of Co3O4 nanoplates in colorimetric sensing technologies should be addressed in detail.

Response 2: We would like to thank this reviewer’s kind comment. 2D nanoparticles are defined as particles with one dimension that is confined to the nanometer length scale (<100 nm). Due to their unique shape, they have high surface-to-volume ratios as well as anisotropic physical and chemical properties compared to 3D nanomaterials (please refer to lines 23-25 in page 2). Through adsorption for dissolved oxygen on the surface, the Co3O4 nanoplates are capable of catalyzing the dissolved oxygen to release reactive oxygen species accompanied by the oxidation of Co2+ to Co3+. In various reactive oxygen species, superoxide anions (O2·−) acted as the main reactive oxygen species for the oxidation of TMB to blue ox-TMB (Zhao, Q. et al. J. Hazard. Mater. 2021, 409, 125019). Due to a larger surface area, Co3O4-200 nanoplates can adsorb and catalyze more dissolved oxygen to release more superoxide anions, showing stronger catalytic activity for the oxidation of TMB (Please refer to lines 21-28 in page 5).

Comment 3. In Figure 2, SEM images of Co3O4 nanoplates seems to be more porous structures with increasing calcination temperature. Author should address the reason why their specific surface areas were decreased at higher calcination temperatures even though they have more porous structures.

Response 3: We would like to thank this reviewer’s kind comment. The SEM image in Fig. 2d showed that the Co3O4-800 presented a macroporous structure due to the fusion and recrystallization of samples in the process of high calcination temperature. The process of fusion and recrystallization also caused the agglomeration, shown in the insert SEM image in Fig. 2d. Therefore, the sample Co3O4-800 has a lower surface area by the comprehensive analysis combined with agglomeration and uneven macropores. A loose surface structure with cracks in Co3O4-200 possesses a high surface area. (Please refer to line 24 in page 4 to line 2 in pages 5)

Comment 4. According to the calcination temperature, the oxidase activities of Co3O4 nanoplates were seriously decreased. Authors should explain this phenomenon and mechanism as well.

Response 4: We would like to thank this reviewer’s kind comment. For the synthesis of Co3O4 by thermal decomposition, the different calcination temperature formed different degrees of crystallinity and surface morphologies, which contributed to the differences in surface area. Co3O4-200 with a larger surface area could absorb more dissolved oxygen to release more reactive oxygen species. Therefore, Co3O4-200 exhibited stronger oxidase-like activity than the Co3O4-500 and Co3O4-800. (Please refer to line 24 in page 4 to line 2 in pages 5)

Thank you very much for many appropriate and valuable comments. I am sure that these comments improved significantly the quality of the manuscript.

Sincerely yours,

Jingjing Du

Reviewer 2 Report

The manuscript studied the dopamine on determination in presence of Co3O4 derived by different calcination temperature.  The structure and catalytic mechanism of the obtained catalysts is discussed in this paper.  I have some comments.

1.     Table 1, the specific area measured is quite different in all three samples. Why it so? Moreover, SEM shows the sample prepared at high temperature has more porous structure. Such samples should show high surface area. The added reason seems not fully justifying results.

2.     What about buffer role? Why acetate not any other buffer?

3.     Was this catalyst reusable? If so, Author should add SEM/TEM image after catalysis to show stability of catalysts. Moreover, what about re-usability of the catalyst?

4.     Did author tried to measure dopamine from real sample?

Author Response

Dear Reviewer:

Many thanks for the comments dated Aug. 19th, 2022 on the manuscript “Synthesis of Co3O4 nanoplates by thermal decomposition for colorimetric detection of dopamine”. We have carefully considered the reviewer’s comments and revised our manuscript accordingly. The changes are highlighted with a yellow pen in the revised manuscript. Here we would like to address the editor’s comments:

Comment 1. Table 1, the specific area measured is quite different in all three samples. Why it so? Moreover, SEM shows the sample prepared at high temperature has more porous structure. Such samples should show high surface area. The added reason seems not fully justifying results.

Response 1: We would like to thank this reviewer’s kind comment. The SEM image in Fig. 2d showed that the Co3O4-800 presented a macroporous structure due to the fusion and recrystallization of samples in the process of high calcination temperature. The process of fusion and recrystallization would cause the agglomeration, shown in the insert SEM image in Fig. 2d. The distribution of pores and pore size was not uniform. Therefore, the sample Co3O4-800 has a lower surface area by the comprehensive analysis combined with agglomeration and uneven macropores. A loose surface structure with cracks in Co3O4-200 possesses a high surface area. (Please refer to line 24 in page 4 to line 2 in pages 5)

Comment 2. What about buffer role? Why acetate not any other buffer?

Response 2: We would like to thank this reviewer’s kind comment. The buffer solution is used to maintain the stability of pH value in a certain range. The activities of mimics enzymes and enzymes are dependent on the pH as well as temperature. The influences of pH and temperature on the activity of Co3O4 here have been discussed. Therefore, the results illustrate that the Co3O4 can keep a high oxidase-like activity in a wide temperature range from 20 to 60 oC with a pH of around 4.0. The acetic acid buffer is the most commonly used buffer for pH=4. (Please refer to lines 16-26 in page 6, Fig. 4)

Fig. X1 Effect of temperature (a, b) and pH (c, d) on the oxidase-like activity of Co3O4-200.

Comment 3. Was this catalyst reusable? If so, Author should add SEM/TEM image after catalysis to show stability of catalysts. Moreover, what about re-usability of the catalyst?

Response 3: We would like to thank this reviewer’s kind comment. At first, We thought that the Co3O4 with a little large size can be easily recycled. However, most black powders were dissolved into the solution during the process of testing. It was reported that the release of reactive oxygen species from dissolved oxygen triggered by Co3O4 is accompanied by the oxidation of Co2+ into Co3+, which causes the dissolution of Co3O4 (Zhao, Q. et al. J. Hazard. Mater. 2021, 409, 125019). Therefore, the catalyst Co3O4 is not reusable. There is no additional SEM/TEM image for recycled Co3O4 sample.

Comment 4. Did author tried to measure dopamine from real sample?

Response 4: We would like to thank this reviewer’s kind comment. Because of the limitation of experimental conditions, the real sample still cannot be tested. In the following research work, we will focus on solving the reusability of catalysts and exploring their practicability.

Thank you very much for many appropriate and valuable comments. I am sure that these comments improved significantly the quality of the manuscript.

Sincerely yours,

Jingjing Du

Round 2

Reviewer 1 Report

The authors have addressed my comments accordingly. It is recommended to publish the manuscript at its current version.

Reviewer 2 Report

Author response to my comments are satisfactory.